# Stair-Climbing Tests or Self-Reported Functional Capacity for Preoperative Pulmonary Risk Assessment in Patients with Known or Suspected COPD—A Prospective Observational Study

**DOI:** 10.3390/jcm12134180

**Published:** 2023-06-21

**Authors:** André Dankert, Benedikt Neumann-Schirmbeck, Thorsten Dohrmann, Lili Plümer, Viktor Alexander Wünsch, Phillip Brenya Sasu, Susanne Sehner, Christian Zöllner, Martin Petzoldt

**Affiliations:** 1Department of Anesthesiology, Center of Anesthesiology and Intensive Care Medicine, University Medical Center Hamburg-Eppendorf, Martinistrasse 52, 20246 Hamburg, Germany; bene.neumann@googlemail.com (B.N.-S.); t.dohrmann@uke.de (T.D.); l.pluemer@uke.de (L.P.); v.wuensch@uke.de (V.A.W.); p.sasu@uke.de (P.B.S.); c.zoellner@uke.de (C.Z.); m.petzoldt@uke.de (M.P.); 2Institute of Medical Biometry and Epidemiology, University Medical Center Hamburg-Eppendorf, 20246 Hamburg, Germany; s.sehner@uke.de

**Keywords:** chronic obstructive pulmonary disease, postoperative pulmonary complication, preoperative, self-reported poor functional capacity, stair-climbing test, pulmonary risk

## Abstract

Background: This prospective study aims to determine whether preoperative stair-climbing tests (SCT) predict postoperative pulmonary complications (PPC) better than self-reported poor functional capacity (SR_PFC_) in patients with known or suspected COPD. Methods: A total of 320 patients undergoing scheduled for major non-cardiac surgery, 240 with verified COPD and 80 with GOLD key indicators but disproved COPD, underwent preoperative SR_PFC_ and SCT and were analyzed. Least absolute shrinkage and selection operator (LASSO) regression was used for variable selection. Two multivariable regression models were fitted, the SR_PFC_ model (baseline variables such as sociodemographic, surgical and procedural characteristics, medical preconditions, and GOLD key indicators plus SR_PFC_) and the SCT model (baseline variables plus SCT_PFC_). Results: Within all stair-climbing variables, LASSO exclusively selected self-reported poor functional capacity. The cross-validated area under the receiver operating characteristic curve with bias-corrected bootstrapping 95% confidence interval (95% CI) did not differ between the SR_PFC_ and SCT models (0.71; 0.65–0.77 for both models). SR_PFC_ was an independent risk factor (adjusted odds ratio (OR) 5.45; 95% CI 1.04–28.60; *p* = 0.045 in the SR_PFC_ model) but SCT_PFC_ was not (adjusted OR 3.78; 95% CI 0.87–16.34; *p* = 0.075 in the SCT model). Conclusions: Our findings indicate that preoperative SR_PFC_ adequately predicts PPC while additional preoperative SCTs are dispensable in patients with known or suspected COPD.

## 1. Introduction

More than 320 million patients undergo surgery each year worldwide [1]. It has been recognized that quantifying functional capacity or cardiopulmonary fitness is a pivotal step for preoperative cardiac risk assessment [2,3,4,5,6]. It has been reported that self-reported functional capacity less than two flights of stairs might improve preoperative cardiovascular risk classification [6]. Current guidelines recommend assessing the self-reported ability to climb two flights of stairs in patients referred for intermediate- or high-risk non-cardiac surgery [3].

Postoperative pulmonary complications (PPC) frequently occur after major surgery [7,8] and the prevalence is particularly high in patients with chronic obstructive pulmonary disease (COPD) [9,10,11,12]. There is growing evidence that individuals with COPD undergoing general anesthesia are at high risk for PPC [10,11]. The incidence of PPC is almost doubled in patients with COPD and the assumed underlying mechanism might be impaired mucociliary clearance of aspirated bacteria and impaired gas exchange [11,12,13]. However, in contrast to cardiovascular risk assessment, only very limited data exist regarding the ability of poor functional capacity to predict PPC after non-thoracic surgery [7,14,15,16,17,18,19,20]. While the prevalence of COPD is rising in the ageing society, COPD is still underdiagnosed and undertreated [21,22]. Assessment of functional capacity, for example with a stair-climbing test or six-minute walk test, is standard of care for staging and monitoring of disease progression in COPD [23,24]. Further, stair-climbing tests have a proven benefit for prediction of PPC after lung resection [25,26,27,28,29,30] and it has been proposed that they might play a relevant role for the prediction of PPC in patients undergoing non-thoracic surgery as well [7,14,15,16,17,18,19,20].

However, screening of surgical patients with stair-climbing tests is costly, labor intensive, and time consuming; it requires adequate infrastructural preconditions and health-care resources and may contribute to additional delays [31]. In contrast, self-reported poor functional capacity is easy to assess without additional expenses and efforts as it is already part of the preoperative routine in many regions and institutions [5,6,32]. However, the diagnostic value of self-reported poor functional capacity for the prediction of PPC is unknown.

This prospective study aims to determine whether preoperative stair-climbing tests predict PPC better than self-reported poor functional capacity in patients with known or suspected COPD undergoing major non-cardiac surgery.

## 2. Materials and Methods

The Preoperative Diagnostic Tests for Pulmonary Risk Assessment in Chronic Obstructive Pulmonary Disease (PREDICT) study is a prospective observational single-center study conducted in accordance with the Declaration of Helsinki. The study design, conduction, and reporting were carried out in accordance with the TRIPOD statement [33]. The study was registered with Clinical-Trials.gov (NCT02566343) and approved by the Ethics Committee of the Medical Association of Hamburg (PV4743, 5 August 2014). Participants gave written informed consent. The present findings result from an analysis of an independent dataset within the PREDICT study as outlined in the study protocol [10].

### 2.1. Patient Allocation and Data Collection

Adult patients who presented at our anesthesia preoperative assessment clinic between 18 November 2014 and 14 July 2016 were assessed for eligibility. Patients with GOLD key indicators for COPD [23] and a COPD assessment test (CAT^TM^) [34] score ≥ 5 points, who were scheduled for major non-cardiac surgery, underwent a structured pulmonary risk stratification including preoperative spirometry. Spirometry was performed before and after bronchodilator application (200 µg salbutamol via a spacer) using a Spirobank G^TM^ spirometer (Medical International Research, Rome, Italy; software: winspiroPRO^TM^, version 3.2) according to the recommended standards [23,35]. Reference values were taken from a Caucasian population [10,36].

Inclusion criteria were elective major non-cardiac surgery defined as an expected mortality >2% [37] with an anticipated operation duration ≥120 min and/or planned postoperative intensive care unit admission, planned general anesthesia, and an expected postoperative hospital stay of at least three days. Patients without GOLD key indicators (dyspnea, chronic cough, chronic sputum production, recurrent lower respiratory tract infections, or smoking history), patients with tracheostoma or planned tracheostomy, pregnant women, patients <18 years, and patients with an inability or non-compliance for spirometry or for assessment of functional capacity were excluded. Patients with confirmed COPD (forced expiratory volume in 1 s/forced vital capacity ≤0.7) were included in the COPD cohort and compared with a reference cohort that included 80 consecutive patients with GOLD key indicators and a CAT score > 5 points in whom spirometry disproved COPD (Figure 1). Self-reported poor functional capacity was assessed and a preoperative stair-climbing test was performed in all participators during the preoperative assessment visit of the patient in the preoperative assessment clinic prior to elective surgery.

### 2.2. Self-Reported Poor Functional Capacity (SR_PFC_)

Within a structured routine preoperative interview patients were asked if they were able to climb two flights of stairs. If the answer was ‘no’, this was documented as self-reported poor functional capacity less than two flights of stairs [6].

### 2.3. Stair-Climbing Test (SCT)

All patients underwent a standardized preoperative stair-climbing test supervised and recorded by a member of the study team (observer). The stair-climbing test was performed in a sparsely frequented stairwell to ensure uniform, reproducible conditions for all patients. After at least 10 min of rest, patients were asked to climb up the stairwell with an individualized rapid pace. Patients were instructed not to skip single steps and to only use the railing to keep balance. The observer recorded the time from the start of the stair-climbing test until the patient reached the final landing or terminated the stair-climbing test prematurely due to symptoms (for example dyspnea, exhaustion, dizziness, or chest pain). The observer followed the patient at a little distance in order to avoid making the pace. The steps of the staircase were 17 cm high and 21 cm deep. The test track had a total height of 12.07 m. This stairwell comprised 8 flights of stairs with a landing between each flight that could easily be passed with two or three additional steps.

### 2.4. Outcome Data

All patients were followed-up until hospital discharge and all available outcome data were recorded. The clinical information systems Soarian Health Archive, release 3.04 SP12 (Siemens Healthcare), critical care information management system (ICM, version 8.12, Draeger Medical), and anesthesia charts were systematically screened for adverse events, newly developed clinical signs, new diagnoses, and new radiological findings. Study inclusion was blinded to all health-care professionals in the operation theater, intensive care unit, and normal ward, and intra- and postoperative managements were left to the discretion of the handling physicians.

In addition, a structured postoperative follow-up was conducted between the first and fifth day after extubation.

### 2.5. Sample Size

The approach of Riley and coauthors [38] was used to calculate the required sample size for model development; it uses three criteria: The sample size should ensure an accurate estimate of the overall outcome risk. We assumed a PPC rate of 65% [9,11,39] and a margin of error ≤0.05. We further claimed that the sample size should lead to a shrinkage of predictor effects of 10% and small optimism in the apparent model fit. A Cox–Snell R^2^ of 0.8 and 8 candidate predictors were assumed to be appropriate. Based on this assumption, a required sample size of 350 patients was approximated. Assuming a dropout rate of 5%, 365 patients were included in our study, and datasets from 320 patients were analyzed, leading to an acceptable relaxation of the restricting margin of error from 0.05 to 0.052 [38].

### 2.6. Primary Endpoint

The primary endpoint of this analysis was PPC, as defined by the European Perioperative Clinical Outcome (EPCO) definitions, as a composite of respiratory infection, respiratory failure, pleural effusion, bronchospasm, atelectasis, pneumothorax, and/or aspiration pneumonitis [40].

### 2.7. Covariables for Model Fitting

Potentially eligible covariables were screened by literature research, previous studies [6,14,15,17,18,25,26,27,28,29,30,34,41,42,43,44], and clinical considerations.

Fixed covariables for both models (baseline variables):Sociodemographic data: age (years), body mass index (kg m^−2^), and sex (male);American Society of Anesthesiologists physical status (ASA I-IV);Cardiovascular diseases (yes/no): any reported cardiovascular precondition;Cancer operation (yes/no): patient scheduled for any kind of tumor operation;Type of surgery: clustered variable with five distinct groups (lower abdominal or orthopedic; ear, nose, and throat; neurosurgery; upper abdominal; thorax and mediastinal);GOLD key indicators: chronic cough (yes/no), sputum production (yes/no), dyspnea (yes/no), recurrent lower respiratory tract infections (yes/no), ever-smoked (yes/no);Pack years (if ever-smoked).

Additional stair-climbing covariables (candidates):Self-reported poor exercise capacity (SR_PFC_) (yes/no): self-reported inability to climb two flights of stairs;Poor exercise capacity in the stair-climbing test (SCT_PFC_) (yes/no): actually, observed inability to climb two flights of stairs in the stair-climbing test;Successful completion of the stair-climbing test (yes/no): patient reached the final landing after climbing 8 flights of stairs;High (m): maximal height reached by the patient (maximum 12.07; final landing);Power (w): calculated as the bodyweight (kg) × height (m) × 9.81 m/s^2^;Pace (stairs/s): average speed of climbing.

### 2.8. Multivariable Model Development and Performance

To evaluate the incremental diagnostic value of the self-reported poor exercise capacity and the stair-climbing test we fitted two multivariable logistic regression models, the ‘SR_PFC_ model’ and the ‘SCT model’, and compared their ability to predict PPC in patients with GOLD key indicators for COPD undergoing major surgery. Complete case analysis was used for both models. Skewed predictors were logarithmized. The results are reported as odds ratios (OR) (95% confidence interval (CI)).

Both models include fixed baseline variables: sociodemographic data, medical preconditions, surgical covariables, and COPD-specific assessments (GOLD key indicators and pack years); these variables were included into the model without any further preselection as they already represent an established assessment bundle [10,23].

STATA’s adaptive least absolute shrinkage selector operator (LASSO) regression, which performs multiple LASSO regression analyses, each with cross validation, was used to select eligible variables out of the six ‘stair-climbing variables’ on top of the fixed baseline variables. This approach was chosen for variable selection and to correct for overfitting by shrinkage. A covariable was considered relevant for prediction if the β-coefficient was not shrunk to zero.

In the next step, we fitted the ‘SR_PFC_ model’, that incorporates self-reported poor exercise capacity in addition to the baseline variables, and the ‘SCT model’, that incorporates SCT_PFC_ in addition to the baseline variables.

To determine the predictive performance, the average of the ten-fold cross-validated area under the receiver operating characteristic (ROC) curve (cvAUC) was calculated with bias-corrected bootstrapped (bc-b) 95% CI for both models.

### 2.9. Descriptive Statistics

Sample characteristics are given as absolute and relative frequencies or mean (standard deviation) as well as median (interquartile range), whichever is appropriate. Differences between COPD and non-COPD patients were compared using Fisher’s exact test, Student’s *t*-test, or the Mann–Whitney U test, whichever was appropriate. A two-tailed *p* < 0.05 was considered statistically significant. We report nominal *p*-values without correction for multiplicity. Statistical analyses were performed using STATA, version 17.0 (StataCorp, College Station, TX, USA).

## 3. Results

A total of 31,714 patients were screened and 1271 individuals with GOLD key indicators received spirometry. A total of 365 patients were included in the PREDICT trial and 320 patients (240 with verified COPD and 80 with negative spirometry) were analyzed (12% drop out rate) (Figure 1) [10]. The dataset of this analysis is complete without missing values. Patients with confirmed COPD were grouped into COPD severity classes based on FEV1% values (78 GOLD I, 125 GOLD II, 28 GOLD III, and 9 GOLD IV) [10]. The baseline characteristics of the COPD and reference cohorts are given in Table 1.

Patients with confirmed COPD more frequently terminated the stair-climbing test prematurely before reaching the final landing, spent less power, and reached a slower pace than patients in the reference cohort without COPD (Table 1). Of the analyzed patients, 65.6% (210/320) developed PPC, 47.5% in the non-COPD reference cohort and 71.7% in the COPD cohort (*p* < 0.001).

### Multivariable Model Development and Performance

No missing values exist and all 320 analyzable datasets were used for model fitting. 

STATA’s adaptive LASSO regression was applied to select eligible stair-climbing variables on top of the baseline variables and only selected self-reported poor exercise capacity (shrunken β-coefficient 3.39) while the β-coefficients of all other five stair-climbing candidate predictors were shrunk to zero (Table 2).

Two multivariable regression models were fitted, the ‘SR_PFC_ model’ (baseline variables plus self-reported poor exercise capacity) and the ‘SCT model’ (baseline variables plus SCT_PFC_). Interestingly, while self-reported poor exercise capacity was an independent risk factor in the ‘SR_PFC_ model’ (adjusted OR 5.45; 95% CI 1.04–28.60; *p* = 0.045), SCT_PFC_ was not an independent risk factor in the ‘SCT model’ (adjusted OR 3.78; 95% CI 0.87–16.34; *p* = 0.075). Sex (male), upper abdominal, and thoracic and mediastinal surgery were independent predictors for PPC in both models (Table 3). 

The cvAUC (bc-b 95% CI) did not differ between the ‘SR_PFC_ model’ (0.712; 0.648–0.768) and the ‘SCT model’ (0.713; 0.650–0.770) (Figure 2).

## 4. Discussion

In this secondary analysis of a prospective observational study in patients undergoing major non-cardiac surgery, we screened a large surgical population with more than 30,000 patients to identify high-risk individuals. Exercise testing was performed in 365 individuals with known or suspected COPD. 

Interestingly, our analysis demonstrates that actually performing a stair-climbing test does not translate into a better diagnostic performance than simply asking the patient if he or she is able to climb two flights of stairs. Only few patients were unable to climb two flights of stairs. Those who reported not being able to climb two flights of stairs were at high risk for PPC. Within all stair-climbing parameters only self-reported poor functional capacity was selected by the LASSO regression. Self-reported poor functional capacity was an independent risk factor associated with a more than five-fold increased risk for PPC, while poor functional capacity in the stair-climbing test was not an independent predictor. Beyond this, only the type of surgery and male sex were further independent risk factors.

The cross-validated area under the receiver operating characteristic curve with bias-corrected bootstrapping 95% confidence interval (95% CI) did not differ between the self-reported poor functional capacity and stair-climbing test models (0.71; 0.65–0.77 for both). Hence, our data indicate that preoperative self-reported poor functional capacity already adequately predicts PPC while preoperative stair-climbing tests did not further improve preoperative pulmonary risk assessment in patients with known or suspected COPD.

Poor functional capacity can either originate from cardiovascular comorbidities (for example, congestive heart failure) or dysfunction of the respiratory system, particularly in COPD. Functional capacity or cardiopulmonary fitness is an important component of preoperative risk assessment [3,4,5,6]. 

Guidelines recommend using self-reported functional capacity for preoperative cardiovascular risk assessment [3]. Recently, a large multicenter study demonstrated that self-reported functional capacity did not improve prediction of major adverse cardiovascular events compared with clinical factors [32]. While functional capacity is an established predictor for major adverse cardiac events [3,4,5,6], still little is known about its role for prediction of PPC [10,44].

It is an established part of the preoperative routine assessment in many institutions to screen functional capacity by posing a very simple question: ‘Can you climb two flights of stairs?’ [3,6,44]. It was unknown if patients that answer ‘no’ are at increased risk for PPC or if more sophisticated diagnostic measures such as stair-climbing tests or a six-minute walk test are required for this purpose [14,17,23,24,26,27,28,29,30]? Unfortunately, preoperative spirometry and blood gas analysis do not improve preoperative prediction of PPC in patients with known or suspected COPD [10]. 

The prevalence of COPD is rising rapidly and the worldwide mean prevalence has been estimated to be 13.1% [45]. While it has been recognized that COPD has great implications on patient outcome in perioperative medicine [9,11,46], still only very few studies have investigated preoperative pulmonary risk assessment in individuals with COPD [9].

Considering all this, the present study intended to evaluate if a structured quantitative assessment of the functional capacity by means of a stair-climbing test predicts PPC more accurately than a simple self-assessment of poor functional capacity gathered during preoperative consultation.

Stair-climbing tests are cumbersome, time consuming, costly, and inconvenient for the patient, as they might provoke symptoms such as dyspnea, exhaustion, dizziness, or chest pain. Moreover, stairwells are not uniform and significantly differ between buildings, regions, or countries, thus, test results are not interchangeable [19]. 

On the other hand, assessment of self-reported poor functional capacity preserves time, costs, and human and health-care resources. Stair-climbing tests are unfeasible in individuals with fractures of the lower limbs, certain types of disabilities, neuromuscular or rheumatic diseases or pain syndromes; however, in many of these individuals, self-reports might still be valuable. Hence, for practical reasons self-assessments might be more feasible and reproducible than exercise tests in many patients.

Beyond all this, the question remains if prediction of the PPC changes perioperative management, decision making, and finally, postoperative outcome [47]? Currently, pulmonary risk prediction tests are neither linked to specific preventive or therapeutic concepts nor provide clear therapeutic targets. Furthermore, efficiency of measures to reduce perioperative risk and to optimize the care in the perioperative period in patients with COPD, such as antiobstructive medication, preoperative pulmonary rehabilitation, choice of drugs, monitoring, or postoperative care is vague [46]. However, the predicted risk for PPC can be particularly useful for shared decision making before surgery. In this context, the estimated risk for PPC might contribute to the individual decision making to undergo surgery or not. Here, the key question is if the benefit of surgery outweighs the combined risk of surgery and anesthesia [47]?

This study has some limitations. Our data represent a single-center experience and caution should be taken extrapolating them to other institutions or different patient populations. Even though we performed a structured postoperative follow-up, assessment of postoperative complications was largely based on chart review. Further external validation could reinforce our findings.

## 5. Conclusions

COPD is associated with a high incidence of PPC. Preoperative pulmonary risk prediction is poorly defined and little is known about the role of functional capacity in these patients. Our findings demonstrate that a standard preoperative pulmonary risk assessment that includes sociodemographic data, medical history, medical preconditions, type of surgery, COPD-specific assessments, and self-reported poor functional capacity already sufficiently predicts PPC in patients with known or suspected COPD. Within all stair-climbing parameters, LASSO regression exclusively selected self-reported poor functional capacity. Self-reported poor exercise capacity was an independent predictor for PPC in the final multivariable model but not poor exercise capacity assessed by stair-climbing tests. In addition to the standard clinical assessment, preoperative stair-climbing tests did not achieve better diagnostic performance than a simple self-report of a poor functional capacity in these patients and are therefore dispensable. The time, costs, human, and health-care resources for a stair-climbing test could be better spent.

## Figures and Tables

**Figure 1 jcm-12-04180-f001:**
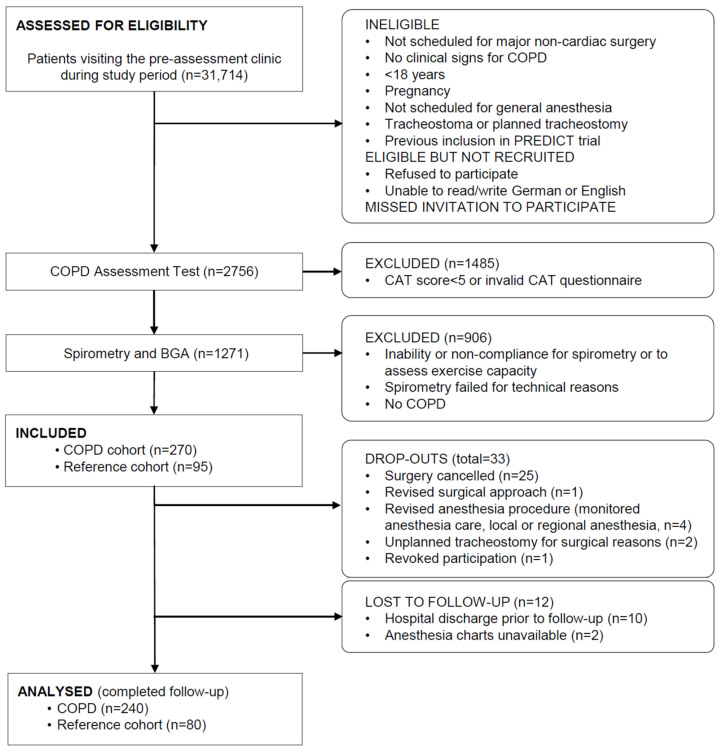
Study flow. Abbreviations: CAT: chronic obstructive pulmonary disease assessment test; COPD: chronic obstructive pulmonary disease.

**Figure 2 jcm-12-04180-f002:**
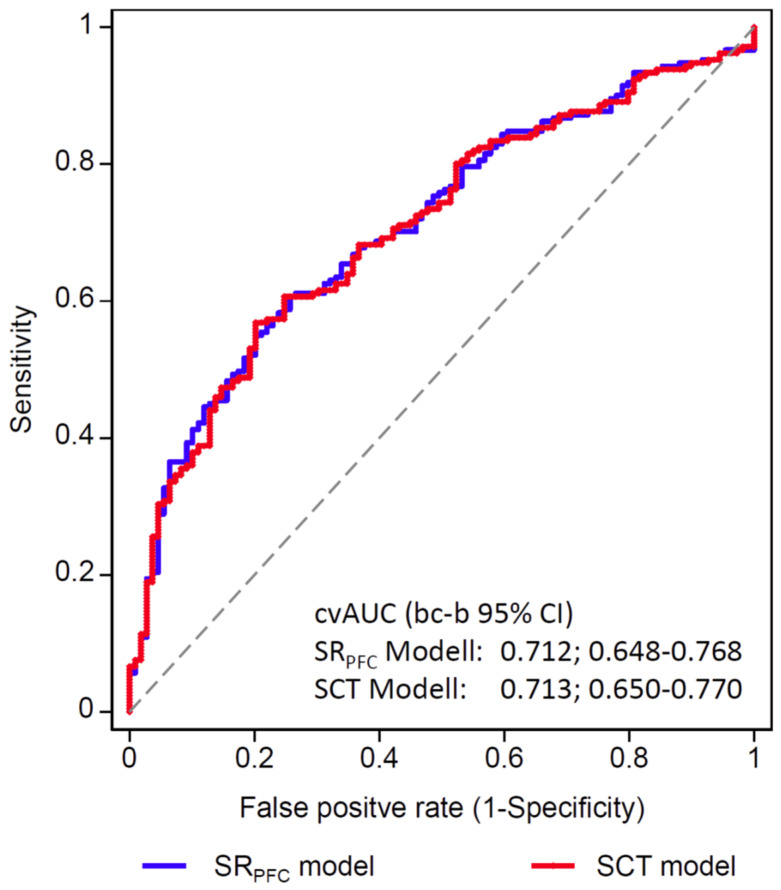
Receiver operating characteristic (ROC) curves and ten-fold cross-validated areas under the ROC curve (cvAUC) illustrate the discriminatory capability of two multivariable models to predict postoperative pulmonary complications in patients with known or suspected COPD undergoing major surgery. Abbreviations: bc-b 95% CI: bias-corrected bootstrapped 95% confidence interval; SR_PFC_: self-reported poor exercise capacity; SCT: stair-climbing test.

**Table 1 jcm-12-04180-t001:** Baseline characteristics.

Variables	Non-COPD (*n* = 80)	COPD (*n* = 240)	*p*-Value
Sociodemographic data			
Age (years)	57.74 ± 13.57	66.25 ± 9.94	<0.001 ^#^
Body mass index (kg m^−2^)	27.17 ± 6.10	25.70 ± 5.87	0.056 ^#^
Sex (male)	42 (52.50)	172 (71.67)	0.002 *
COPD severity grades (GOLD)			
I	-	78 (32.50)	-
II	-	125 (52.08)	-
III	-	28 (11.67)	-
IV	-	9 (3.75)	-
Comorbidities and preconditions			
Bronchial asthma	12 (15.00)	29 (12.08)	0.562 *
Obstructive sleep apnea syndrome	5 (6.25)	16 (6.67)	1.000 *
Coronary artery disease	14 (17.50)	62 (25.83)	0.172 *
Congestive heart failure	1 (1.25)	38 (15.83)	<0.001 *
Diabetes mellitus	10 (12.50)	49 (20.42)	0.135 *
Peripheral arterial occlusive disease	5 (6.25)	35 (14.58)	0.053 *
Renal failure	11 (13.93)	53 (22.08)	0.145 *
Smoking status			
Ever-smoked	62 (77.50)	232 (96.67)	<0.001 **
Pack years	9.88 (0.00–102.00)	40.00 (0.00-189.00)	<0.001 **
Stair-climbing			
Self-reported inability to climb two flights (y/n)	1 (1.25)	18 (7.50)	0.052 *
Stair-climbing test			
Unable to climb two flights of stairs (y/n)	2 (2.50)	21 (8.75)	0.079 *
Successful completed (y/n)	59 (73.75)	128 (53.33)	0.002 *
High (m)	12.07 (11.26–12.07)	12.07 (6.80–12.07)	0.001 **
Power (W)	188.59 ± 71.18	146.14 ± 65.06	<0.001 **
Pace (stairs/sec)	1.37 ± 0.40	1.12 ± 0.40	<0.001 **

The dataset of this analysis is complete without missing values; values are mean ± standard deviation, median (IQR), or number (proportion), whichever is appropriate; differences between COPD and non-COPD patients were compared using Fisher’s exact test *, Student’s *t*-test ^#^, or the Mann–Whitney U test **, whichever was appropriate. Abbreviations: GOLD: global initiative for chronic obstructive lung diseases.

**Table 2 jcm-12-04180-t002:** Adaptive LASSO regression for the selection of eligible stair-climbing covariables out of six candidate predictors on top of the baseline variables.

Covariables	β-Coefficients with λ_min_ = 0.00054
Stair-climbing variables	
Self-reported inability to climb 2 flights of stairs (SR_PFC_) (y/n)	3.39
Observed inability to climb 2 flights of stairs in the SCT (SCT_PFC_) (y/n)	Shrunk to zero
Successful completion (y/n)	Shrunk to zero
High (m)	Shrunk to zero
Power (W)	Shrunk to zero
Pace (stairs/sec)	Shrunk to zero
Baseline variables (fixed)	
COPD-specific assessments (GOLD key indicators and smoking history)	
Chronic cough (y/n)	1.26
Sputum production (y/n)	1.70
Dyspnea (y/n)	0.97
Recurrent lower respiratory tract infections (y/n)	2.06
Ever-smoked (y/n)	0.83
Pack years (if ever-smoked) ^1^	1.14
Preconditions, sociodemographic, surgical, and procedural characteristics	
Age (years)	1.02
Body mass index (kg m^−2^) ^1^	1.17
Sex (male)	2.09
ASA physical status	
ASA I and II	(Reference)
ASA III	0.99
ASA IV	0.936
Cardiovascular diseases (y/n)	1.30
Cancer operation (y/n)	1.09
Type of surgery:	
Lower abdominal or orthopedic	(Reference)
Ear, nose, and throat	0.71
Neurosurgery	1.01
Upper abdominal	3.28
Thoracic and mediastinal	9.27

Reported are λ_min_—penalty parameter determined by adaptive LASSO, β-coefficients of the best fitting LASSO regression model that was not shrunk to zero; covariables were considered informative if coefficients were not shrunk to zero; ^1^ per two-fold increase. The dataset of this analysis is complete without missing values. Abbreviations: SCT: stair-climbing test; SR: self-reported; PFC: poor functional capacity.

**Table 3 jcm-12-04180-t003:** Multivariable logistic regression analysis: fitting of two multivariable prediction models for postoperative pulmonary complications.

	SR_PFC_ Model		SCT Model	
	Adjusted OR (95% CI)	*p*-Value ^2^	Adjusted OR (95% CI)	*p*-Value ^2^
Poor functional capacity (inability to climb two flights of stairs)				
Self-reported (SR_PFC_)	5.45 (1.04–28.60)	0.045	–	
Observed in the stair-climbing test (SCT_PFC_)	–		3.78 (0.87–16.34)	0.075
COPD-specific assessments (GOLD key indicators and pack years)				
Chronic cough	1.17 (0.60–2.28)	0.485	1.17 (0.60–2.28)	0.642
Sputum production	1.59 (0.89–2.82)	0.062	1.74 (0.99–3.07)	0.054
Dyspnea	0.76 (0.40–1.46)	0.862	0.98 (0.53–1.81)	0.951
Recurrent lower respiratory tract infections	1.91 (0.62–5.86)	0.193	1.94 (0.65–5.82)	0.236
Ever-smoked	0.74 (0.20–2.71)	0.806	0.75 (0.21–2.69)	0.656
Pack years (if ever-smoked) ^1^	1.12 (0.90–1.39)	0.180	1.17 (0.94–1.45)	0.162
Sociodemographic, surgical and procedural data, medical preconditions				
Age; years	1.01 (0.99–1.04)	0.257	1.01 (0.99–1.04)	0.291
Body mass index; kg m^−2 1^	1.14 (0.44–2.94)	0.782	1.12 (0.43–2.89)	0.817
Sex (male)	2.17 (1.18–4.00)	0.013	2.12 (1.15–3.91)	0.016
ASA physical status				
ASA II	reference		reference	
ASA III	0.98 (0.51–1.92)	0.963	0.98 (0.51–1.91)	0.961
ASA IV	0.88 (0.26–2.90)	0.828	0.86 (0.26–2.86)	0.802
Cardiovascular diseases	1.31 (0.70–2.44)	0.392	1.26 (0.68–2.33)	0.470
Cancer operation	1.09 (0.62–1.90)	0.762	1.08 (0.62–1.89)	0.780
Type of surgery:				
Lower abdominal or orthopedic	reference		reference	
Ear, nose, and throat	0.71 (0.29–1.77)	0.467	0.72 (0.29–1.79)	0.485
Neurosurgery	1.02 (0.44–2.33)	0.969	1.08 (0.47–2.50)	0.848
Upper abdominal	3.35 (1.60–7.04)	0.001	3.34 (1.59–7.01)	0.001
Thoracic and mediastinal	9.38 (3.33–26.42)	<0.001	9.20 (3.27–25.89)	<0.001

The dataset of this analysis is complete, without missing values; the results are reported as adjusted odds ratios (ORs) with 95% confidence intervals (95% CIs). ^1^ per two-fold increase; ^2^ reported *p*-value results from the Wald χ^2^ test. Abbreviations: ASA: American Society of Anesthesiologists; COPD: chronic obstructive pulmonary disease; GOLD: global initiative for chronic obstructive lung disease, SCT: stair-climbing test; SR: self-reported; PFC: poor functional capacity.

## Data Availability

Not applicable.

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
