# Peer review of "Stair-Climbing Tests or Self-Reported Functional Capacity for Preoperative Pulmonary Risk Assessment in Patients with Known or Suspected COPD—A Prospective Observational Study"

_jcm, 2023, doi:10.3390/jcm12134180_

Round 1

Reviewer 1 Report

I appreciate the opportunity to review this  manuscript. Dankert and colleagues have designed a prospective study determining whether preoperative stair-climbing tests (SCT) predict postoperative pulmonary complications (PPC) better than self-reported poor functional capacity (SRPFC) in patients with known or suspected COPD. They conclude that preoperative SRPFC adequately predicts PPC while additional preoperative SCTs are dispensable in patients with known or suspected COPD. Additional questions and comments about the manuscript are itemized below:

Abstract:

-- Page 8, line 24: Please write out the term for SCTPFC.

 Introduction:

-- Page 2, line 57: "...it is already part of the preoperative routine in many regions and institutions". where is the reference to support this statement?

Materials and Methods:

-- Sample size: Sample size justification is a key feature of the design of a research study. Please explain and justify how the sample size was chosen. For example, what is the basis for assuming that the incidence of PPC is 65%, and is there any literature to support it?

-- Primary endpoint: Please elaborate on how to collect data on postoperative pulmonary complications in patients. The deadline for data collection?

-- Descriptive statistics: Explain how missing data were addressed. What method was used?

 Results:

-- Page 5, line 203: The paper would benefit from a flow diagram of patient disposition. 

Discussion:

-- There needs to be more discussion on why preoperative SRPFC adequately predicts PPC while additional preoperative SCTs are dispensable in patients with known or suspected COPD. The discussion section of the article should focus on the findings in the current work.

 Conclusions:

-- The conclusions section should be more concise and targeted on the main findings of this study.

Reviewer 2 Report

GENERAL COMMENTS

The authors performed a prospective study to determine whether preoperative stair-climbing tests predict PPC better than self-reported poor functional capacity in patients with known or suspected COPD undergoing major non-cardiac surgery.

It is very important to prevent the PPC, so this study provides information about the best measure to know the functional capacity preoperative.    

However, there are some specific comments that it is necessary to resolve.

SPECIFIC COMMENTS

The title should be brief.

INTRODUCTION

What are the PPC more common for COPD patients?

METHODS

It's been a long time since the recruitment of patients, it started 9 years ago.

Line 78-79: “Details have been reported elsewhere”. The inclusion criteria should appear in the manuscript.

Line 75-77: the authors called patients with known or suspected COPD, although, the patients have GOLD key indicators for COPD according to the inclusion criteria.

The inclusion criteria should be clearer, the characteristics of COPD patients influence in the PPC depending on the severity of disease (spirometry, symptoms, etc.)

Line 90-92: when are evaluated it?

Line 196: what are the characteristics of non-COPD patients’ group? Are healthy people?

RESULTS

Table 1. It didn’t include characteristics about the severity of COPD.

Did the authors record the PPC after the surgery?

Round 2

Reviewer 2 Report

The authors revised the manuscript well